# Recent Progress on NIR Fluorescent Probes for Enzymes

**DOI:** 10.3390/molecules27185922

**Published:** 2022-09-12

**Authors:** Jintao Zhao, Tao Ma, Bingbing Chang, Jianguo Fang

**Affiliations:** State Key Laboratory of Applied Organic Chemistry, College of Chemistry and Chemical Engineering, Lanzhou University, Lanzhou 730000, China

**Keywords:** near-infrared, enzyme, fluorescent probe

## Abstract

The majority of diseases’ biomarkers are enzymes, and the regulation of enzymes is fundamental but crucial. Biological system disorders and diseases can result from abnormal enzymatic activity. Given the biological significance of enzymes, researchers have devised a plethora of tools to map the activity of particular enzymes in order to gain insight regarding their function and distribution. Near-infrared (NIR) fluorescence imaging studies on enzymes may help to better understand their roles in living systems due to their natural imaging advantages. We review the NIR fluorescent probe design strategies that have been attempted by researchers to develop NIR fluorescent sensors of enzymes, and these works have provided deep and intuitive insights into the study of enzymes in biological systems. The recent enzyme-activated NIR fluorescent probes and their applications in imaging are summarized, and the prospects and challenges of developing enzyme-activated NIR fluorescent probes are discussed.

## 1. Introduction

Enzymes are significant biocatalysts that play vital roles in a variety of biologically transformations, assisting organisms in controlling the proper physiological processes under a variety of environmental circumstances. It is well recognized that numerous illnesses and abnormalities of physiological processes are closely correlated with the up- or down-regulation of certain enzyme activity [1]. Therefore, the accurate real-time detection of unusual changes in certain enzymes in biological systems would be beneficial. Additionally, quantifying the in-situ substrate synthesis mediated by enzymatic processes is advantageous for biological research, illness diagnosis, treatment, and the assessment of therapeutic benefits. To determine the concentration of enzymes and their activity, biological samples are often homogenized using a variety of techniques, including lysis, digestion, sonication, or high-pressure homogenization [2]. However, the laborious homogenization process completely destroys the integrity of the material being studied and is not appropriate for real-time enzyme activity in their original state. The lack of a complete and user-friendly technique for illustrating enzyme distribution and expression with spatial resolution may lead to more confusion in knowledge of enzyme control in disease biology.

In order to reflect the levels of biological tissue enzyme expression, the researchers used imaging techniques like positron emission tomography (PET), positron emission tomography (PET), single photon emission computed tomography (SPECT), and so on [3,4,5]. Without requiring surgery or invasive biopsies, these in vivo molecular imaging methods define the enzyme levels in the sick area under examination. The methods now employed in medical imaging applications, meanwhile, are pricy, need a long acquisition time, have a limited range of sensitivity, and are unable to discern subcellular features. Instead of using ionizing radiation, optical imaging methods may create affordable, real-time pictures with great sensitivity and spatial resolution. In preclinical research, fluorescence microscopy is frequently used to measure enzyme activity, and as technology develops, it may also be employed in clinical molecular imaging in the future [6,7,8]. Furthermore, because to their simplicity in synthetic modification, small-molecule enzyme probes can locate enzymes with great sensitivity and specificity in their natural environment [9,10,11]. In actuality, the detection of numerous analytes, including metal ions, small molecules, and biomacromolecules, has been extensively done using fluorescence imaging [12,13,14,15,16,17,18,19,20,21,22]. Near-infrared (NIR) emitting (650–900 nm) fluorescent compounds, in particular, are utilized in biological systems because they can prevent autofluorescence, reduce photon scattering, and promote deep tissue penetration [23,24]. NIR fluorescent dyes have acquired considerable support in clinical optical imaging applications due to continual advancements in the field of deep tissue optical microscopy and advances in imaging algorithms [25]. Significant progress has been achieved in the creation of NIR fluorescence probes based on enzyme activation during the last decade, which is unquestionably advantageous to the early detection and treatment of associated disorders [8,23,26,27,28]. As a result, it fuels the ongoing development of novel NIR enzyme probes. In this study, we will look into NIR fluorescent probes with various photophysical characteristics and their use in detecting enzyme activity under biological settings. We briefly discuss one or more typical near-infrared fluorescent probes for each class of enzymes that may be used to observe enzyme activity in biological systems.

## 2. Design Strategies of NIR Fluorescent Probes for Enzymes

Typically, fluorescent probes are made up of a fluorophore, a linker, and a recognition group. The NIR fluorescent probe’s structural makeup is the same, but the crucial distinction is that the fluorophore is a dye that emits in this wavelength range (650–900 nm). The design of fluorescent probes frequently incorporates a variety of fluorophores with NIR emission properties, including boron-dipyrromethane (BODIPY), dicyanomethylene-4*H*pyran, silicon rhodamine, cyanine, and benzoxazine derivatives [19,29,30,31,32,33,34,35]. When the NIR fluorescent probe crosses through the cell membrane and enters the cell, the fluorescence signal is either “always-on” or “turn-on”. The utilization of enzymatic ligands (e.g., inhibitors and peptides) linked to NIR fluorophores is a frequent method in the former. The built fluorescent probe, on the other hand, has unique conformation that enables its fluorescence to be quenched before binding to the target if the spatial distance between the fluorophore and the inhibitor is rationally changed or an environment-sensitive fluorophore is used. When the probe’s recognition group attaches to the enzyme, altering its conformational change produces an increased fluorescence signal. Although the selectivity of the recognition analyte may be greatly increased using this NIR fluorescent probe design strategy, its use is somewhat constrained by the complexity of molecular design. This review’s main focus is on the latter, which is the development trend and construction progress of NIR enzyme-activated fluorescence probes based on certain processes. The switching of PET (photon-induced electron transfer), FRET (Fçrster resonance energy transfer), and ICT (intramolecular charge transfer) processes is frequently cited as the mechanism for the fluorescence change of fluorescent probes [9,36,37]. Additionally, ratio-type fluorescent probes are frequently created when the initial state of the activatable fluorescent probe produces fluorescence itself and following particular recognition with the biological target generates fluorescence distinct from the initial state. Furthermore, differences in enzyme activity at distinct locations within the cell can be reflected by linking chemical units that specifically target organelles to probes.

## 3. NIR Fluorescent Probes for Hydrolases

Hydrolases are a class of enzymes that hydrolyze bigger molecules into smaller ones in the presence of water molecules. They play a direct role in a wide range of biological processes that are crucial for maintaining healthy physiological conditions. For instance, during food digestion, energy production, waste excretion, and macromolecular fragmentation, hydrolases are necessary to break down biopolymers into crucial monomers. Figure 1 lists recently reported representative examples of NIR fluorescent hydrolase probes and Table 1 shows the results of the photophysical properties and kinetics studies.

### 3.1. NIR Fluorescent Probes for Carboxylesterase

Carboxylesterase (CE) is a type of enzyme found in many organisms that can catalyze the hydrolysis of a variety of endogenous and exogenous substrates. In their native host organisms, these enzymes fulfill critical biological tasks. Abnormal CE activity is linked to a variety of illnesses, including obesity, hyperlipidemia, and cancer [58,59]. NIR fluorescent probes have been reported to map CE activity in recent years.

In 2016, Yang et al. reported the NIR fluorescent probe **1** for the highly sensitive detection of CE2 (Figure 1) [38]. The NIR fluorophore is DDAO, and a benzoyl group was introduced into DDAO by esterification. The fluorescence was increased 57-fold after interaction with CE2 due to the release of the fluorophore DDAO (Figure 2A). Furthermore, probe **1** was devised to detect CE2 in vivo for the first time. Sun et al. reported a structurally comparable NIR fluorescent probe **2** to probe **1** for CE2 detection, in which the NIR fluorophore DDAO is replaced with a similar one (Figure 1) [39]. Similarly, probe **2** has a very selective ability to detect CE2. Furthermore, utilizing probe **2**, researchers developed a high-throughput screening approach for CE2 inhibitors and discovered four new CE2 inhibitors from various herbal remedies. Zhang et al. developed a dicyanoisophorone-based NIR fluorescent probe **3** for specifically monitoring CE1, which has an acetoxy group as a recognition unit (Figure 1) [40]. The probe and CE1 co-incubated resulted in an emission peak at 665 nm (λ_ex_ = 535 nm) and a relatively large Stokes shift (130 nm) for probe **3**, making probe **3** greater anti-interference ability, suited for usage in biological systems. Wang et al. recently published an NIR fluorescent probe **4** with a greater stokes shift (160 nm) than probe **3** [60]. It has also been used successfully to track CEs activity in living cells and in vivo.

**Figure 2 molecules-27-05922-f002:**
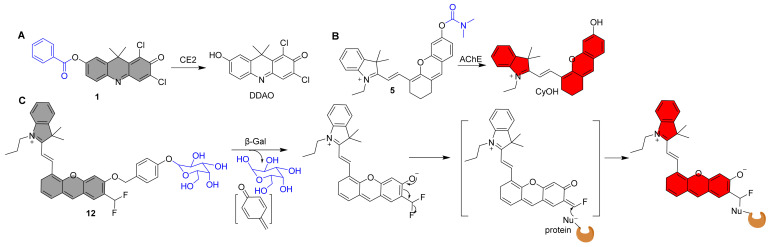
Recognition mechanism of probe **1**, **5**, and **12**.

### 3.2. NIR Fluorescent Probes for Acetylcholinesterase

Acetylcholinesterase (AChE) is an essential hydrolase that controls acetylcholine via hydrolysis, which is an important mechanism in nervous system regulation. Hence, acetylcholinesterase activity is closely associated to depression-like mood disorders. Furthermore, acetylcholinesterase is engaged in a variety of noncholinergic physiological tasks, including cell growth and differentiation, death, and pathogenic processes such as Alzheimer’s disease and carcinogenesis [41,61]. As a result, developing a reliable approach for real-time in situ monitoring of endogenous AChE in vivo is crucial for understanding AChE’s physiological function.

Guo et al. described an NIR fluorescent probe **5** for detecting AchE (Figure 1) [42]. By attaching the carbamyl to the NIR fluorescent fluorophore CyOH, the probe became nonfluorescent. CyOH was produced after interacting with AChE, with a noticeable NIR fluorescence amplification at 700 nm (Figure 2B). Experiments with selectivity reveal that the probe has a strong selectivity for AChE over BChE. Furthermore, the probe may be utilized to image the dynamic changes in AChE activity in PC12 cells in response to retinoic acid and hypoxia. Furthermore, because the probe possesses NIR fluorescence characteristics, it may be used to follow the distribution of intrinsic AChE in zebrafish. Yue et al. created a BODIPY-based NIR fluorescent probe **6** with a dimethyl carbamate group as a recognition unit for monitoring AChE in live cells and mice (Figure 1) [43]. After introducing AChE to the test system, there was an increase in NIR fluorescence at 740 nm. When the probe was incubated with physiologically active species or ions, the fluorescence at 740 nm was not activated, suggesting high selectivity for AChE. Notably, the first visualization demonstrates that dietary restriction (DR) has a significant effect on AChE in the brain of aging mice, which may help provide a new basis for the treatment of various age-related diseases, owing to the probe’s excellent NIR fluorescence optical properties and high selectivity for AChE. Kim et al. recently created a 2-dicyanomethylene-3-cyano-4,5,5-trimethyl-2,5-dihydrofuran-based NIR fluorescent probe **7** for assaying AChE expression in live cells and zebrafish (Figure 1) [44]. It also has a strong selectivity for AChE and the capacity to photograph in vivo in real time.

### 3.3. NIR Fluorescent Probes for β-Galactosidase

β-Galactosidase (β-gal) is a glycosidase subclass whose primary function is to cleave glycosidic linkages on β-galactosides into sugars. Because the activities of these enzymes vary greatly in tumor cells, abnormal glycosylation can be used as a biomarker for cancer detection and treatment. Several β-gal catalyzed in vivo glycosidic bond cleavage based NIR fluorescent probes have been developed in recent years.

Tung et al. published the first NIR fluorescent probe **8** for recognizing β-gal (Figure 1). The probe’s chemical structure is a conjugate of β-galactoside and DDAO, which once reacting with β-gal produces a fluorophore DDAO with an NIR fluorescence emission peak (659 nm). Furthermore, in vivo real-time detection of β-gal activity was accomplished [45].

Photothermal treatment (PTT) has gained popularity in recent years due to its high temporal and spatial controllability and minimally invasive advantages. Pu et al. reported a β-gal-activatable probe **9** with NIR, PA (photoacoustic), and photothermal signaling capabilities for imaging-guided treatment by attaching a β-galactose-caged NIR dye to a PEG chain (Figure 1) [46]. Following cleavage by β-gal, the probe displayed significantly enhanced NIR fluorescence at 720 nm and a PA signal at 680 nm. The combination of PEG chains increases probe solubility, thermal stability, and passive tumor targeting. The highest tumor temperature of mice after probe **9** treatment was 48.38 °C, which was 9.5 °C and 9.38 °C higher than the maximum tumor temperature of the control compound and saline treated animals, respectively. These findings demonstrated that the amount of probe **9** in the tumor was adequate to cause photothermal heating over the cell ablation threshold temperature. Finally, selective PTT effectiveness with little toxicity to normal tissues was obtained.

Zhu et al. described an NIR probe **10** based on BODIPY dye for ratiometric detection of β-gal activity (Figure 1) [47]. After excitation at 530 nm, the probe itself produces fluorescence at 575 nm. NIR fluorescence singal at 730 nm results from a specific reaction with β-gal. Endogenous β-gal activity in cancer cells and tumor-bearing animals can be detected with this probe. Soon after, Ma et al. reported another instance of NIR ratiometric probe **11** being used to detect β-gal (Figure 1) [48]. The fluorescence peak intensity at 580 nm reduced after the addition of β-Gal to the probe solution, but the peak intensity at 650 nm grew dramatically over time. After around 15 min, the F_650_/F_580_ ratio finally equalized and improved by almost 10 folds (0.75 to 7.57). Cui et al. developed a self-immobilizing NIR fluorescent Probe **12** for imaging β-gal (Figure 1) [49]. The probe has a difluoromethyl group in which the hemicyandiamide skeleton serves as an NIR dye and the β-galactose residue serves as a trigger that the enzyme can identify. The probe is triggered by β-Gal to release an electrophilic quinone methyl compound that fluoresces at 709 nm, and it is subsequently grabbed by the target enzyme or a neighboring protein to form covalent bonds, retaining the NIR signals (Figure 2C). The probe showed marked retention in senescence-induced animal models, which provided a wider time window to allow clearance of circulating background fluorescence signals.

Targeting subcellular organelles with fluorescence is essential for investigating particular targets. Liu et al. developed a lysosome-targeting NIR fluorescent probe **13** for detecting β-gal in ovarian cancer cells (Figure 1) [50]. After incubation with β-gal, the probe shows intense NIR fluorescence at 725 nm. Notably, based on time-lapse photos, it may identify β-gal in ovarian cancer cells ultrafast (as quick as 1 min). Because of the probe’s lysosome targeting ability, it may be highly localized in the lysosome, making it more effective than probes without lysosome targeting ability in detecting endogenous β-gal. Luo et al. recently described a β-gal-activatable NIR turn-on fluorescence probe **14** (Figure 1) [51]. The responsive unit of probe **14** was destroyed in the presence of β-gal, exposing the bare phenolic group, resulting in an enhanced fluorescence signal at 770 nm (Figure 1). Finally, NIR fluorescent probe **14** was successfully employed to track endogenous β-gal in tumor-bearing nude mice.

### 3.4. NIR Fluorescent Probes for Phosphatase

Phosphatases are hydrolase components that serve an important regulatory function in phosphate hydrolysis in a range of biomolecules. These enzymes have a role in a variety of physiological processes, including cell development and differentiation, metabolism, and immunology [62]. The study of diseases benefits from the use of NIR fluorescent probes for phosphatase activity in tissues.

Zhang et al. published the first NIR fluorescent probe **15** for the detection of alkaline phosphatase (ALP) in 2017 [52]. It is made up of an NIR fluorophore and monophosphate as the ALP recognition unit (Figure 1). After 30 min of specific reactivity with 40 U/L ALP, the probe exhibits a 57-fold increase in fluorescence at 706 nm. Finally, the probe was found to be effective in detecting ALP in live cells and in vivo. Soon later, using the same design methodology, Nie et al. reported a structurally similar NIR fluorescent probe **16** for ALP detection (Figure 1) [53]. Although probe **16** and probe **15** show similar levels of good ALP selectivity, probe **16**’s maximum emission wavelength is 32 nm redder than probe **15**’s, allowing for improved in vivo imaging. The modified probe **16** may be widely employed in vivo to measure ALP activity, according to results from live cell and animal imaging studies. By combining fluorescent processes into enzyme-activated in situ self-assembly, Ye et al. created an NIR fluorescence and magnetic resonance bimodal probe **17** in 2019 (Figure 1) [54]. When ALP hydrolyzes the phosphate group in the probe, clustered nanoparticles packed on the cell membrane develop, and some nanoparticles can subsequently enter tumor cells via endocytosis. These activities cause NIR fluorescence and magnetic resonance signals in live cells to be amplified. More notably, the bimodal probe **17** was used for noninvasive detection of ALP activity during image-guided surgery in tumor-bearing mice. Hou and co-workers developed a new NIR ratiometric fluorescent probe **18** to measure ALP activity (Figure 1) [55]. The fluorescence peak intensity at 580 nm declined after ALP reaction, but the peak intensity at 650 nm grew dramatically with time. After around 15 min, the F_650_/F_580_ ratio finally equalized and improved by roughly 14 times. The fluorescence characteristics of probe **19** after it reacted with ALP were similar to those of probe **18**, and a ratiometric fluorescence change was also observed (Figure 1) [56]. After approximately 9 min, the F_689_/F_570_ ratio had finally equalized and improved by approximately 4.5 times. Finally, probes **18** and **19** were all successfully employed to scan ALP in vivo, suggesting that they might play an important role in the early detection of ALP-related disorders in the future. Probe **20** exhibits a high selectivity and a low detection limit (0.87 U/L), as well as a substantial Stokes shift (185 nm) (Figure 1) [57]. At the same time, it shows a clear color shift in aqueous solution that is apparent to the human eye. Finally, ALP imaging in mice was accomplished.

## 4. NIR Fluorescent Probes for Oxidoreductases

Oxidoreductases catalyze the transfer of electrons from a donor (reducing agent) to an acceptor (oxidant), typically using cofactors such as nicotinamide adenine dinucleotide phosphate (NADP) or nicotinamide adenine dinucleotide (NAD^+^) [63,64]. The unique roles of oxidoreductases in disease diagnosis, prognosis, therapy, drug metabolism, and various industrial applications have piqued the interest of many researchers. Individual oxidoreductases are involved in the maintenance of specific physiological and biological processes, making them an appealing research target. Figure 3 lists recently reported representative examples of NIR fluorescent oxidoreductase probes and Table 2 shows the results of the photophysical properties and kinetics studies.

**Figure 3 molecules-27-05922-f003:**
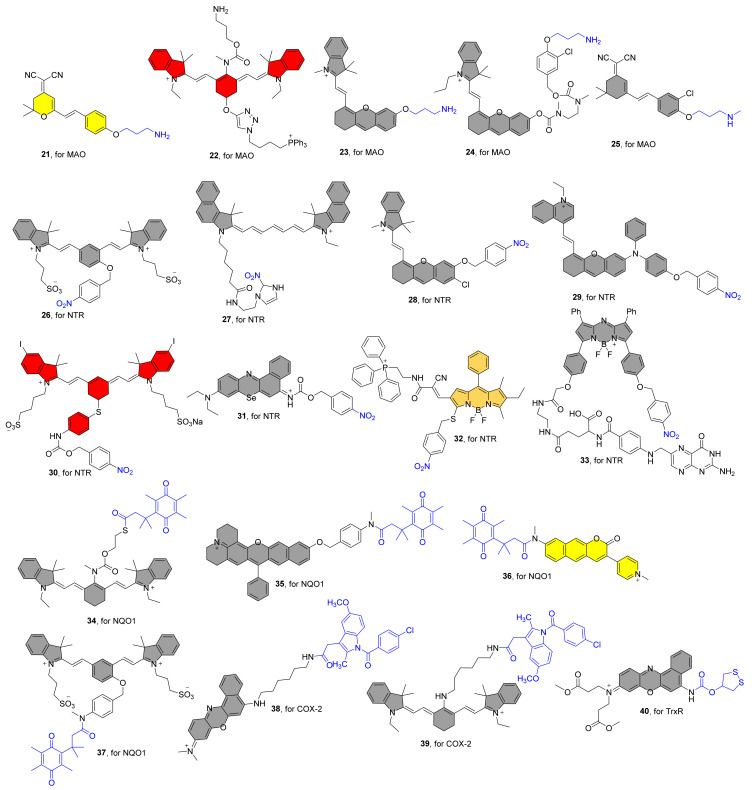
Structures of NIR Fluorescent Probes **21**–**40**.

### 4.1. NIR Fluorescent Probes for Monoamine Oxidases

Based on its substrate and inhibitor selectivity, monoamine oxidase (MAO), a crucial enzyme involved in numerous chemical modifications of neurons to shield them from damage caused by stimuli, is categorized as MAO A and B. In cardiovascular and neurodegenerative diseases, MAOs are overexpressed [85,86]. There have been frequent reports on the use of NIR fluorescent probes to map MAOs activities in biological systems.

Zhang et al. generated the first NIR fluorescent probe **21** for detecting MAO in 2016 (Figure 3) [65]. The 3-amino-propoxy group was incorporated to the NIR fluorophore to construct probe **21**, which reacted with MAO to achieve fluorescence release via a sequential amine oxidation/β-elimination mechanism (Figure 4A). After the addition of MAO, the fluorescence intensity at 550 nm gradually decreased. MAO-B and MAO-A were found to enhance the fluorescence at 664 nm by 7.2 and 3.8 folds, respectively. Finally, in both HeLa and Hep G2 cells, probe **21** was validated to report MAO activity as a change in fluorescence intensity. Probe **22**, which consists of heptamethine cyanine as an NIR fluorophore, 3-amino-propoxy group as a recognition group, and triphenylphosphonium cation as a mitochondrial targeting group, has been reported to detect MAO-B in vivo (Figure 3) [66]. After a specific reaction with MAO-B, the fluorescence spectra exhibited a remarkable decrease at 803 nm to an increase at 750 nm, resulting in a ratiometric fluorescence change (F_750nm_/F_803nm_). The probe was also used to assess MAO-B levels and activity in an H_2_O_2_-induced cellular senescence model and in mice, and the results revealed that elevated MAO-B levels are associated with the aging process. Qin and coworkers also reported an NIR mitochondrial-targeted fluorescent probe **23** for bioimaging and evaluating MAO-A in organisms in another study (Figure 3) [67]. Under test conditions, probe **23** exhibits a 31-fold NIR fluorescence response (λ_em_ = 710 nm) to MAO-A and no response to MAO-B. Colocalization imaging revealed that the probe primarily accumulated in cell mitochondria, enabling for accurate detection of MAO-A activity in subcellular organelles. Finally, the probe was used to measure MAO-A activity in CCl_4_-induced hepatic fibrosis rats, and the results revealed that MAO-A levels were elevated. In contrast to probe **23**, probe **24** has a self-eliminating linker between the recognition site and the fluorophore (Figure 3) [68]. Additionally, probe **24** exhibits strong selectivity for MAO-A over MAO-B, making it a useful tool for figuring out how MAO-A works in cancer because it can be visualized in mouse models. Li and colleagues recently created an NIR fluorescent probe **25** for MAO-A detection that employs propylamine as the recognizing moiety and chlorine-substituted dicyanoisophorone as the NIR fluorophore. (Figure 3) [69]. The fluorescence enhancement caused by MAO-A was 195-fold greater than that caused by MAO-B, indicating high selectivity for MAO-A. Importantly, it can detect MAO-A activity in glucose-deprived H9c2 cardiomyocytes, zebrafish, and ISO-induced failing heart tissue.

**Figure 4 molecules-27-05922-f004:**
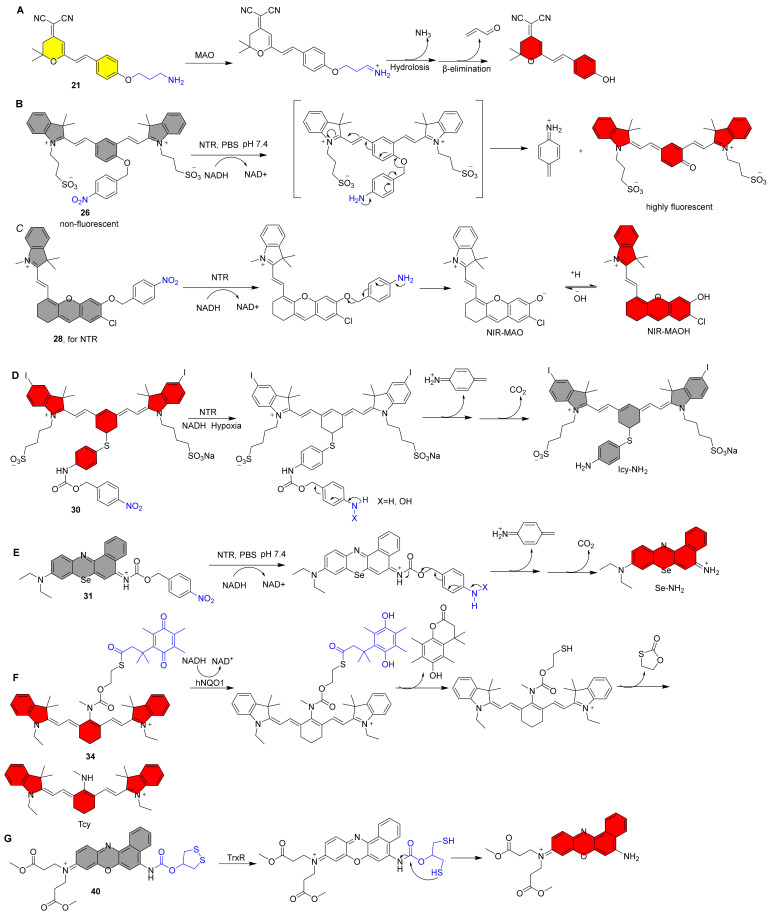
Recognition mechanism of probe **21**, **26**, **28**, **30**, **31**, **34** and **40**.

### 4.2. NIR Fluorescent Probes for Nitroreductase

Nitroreductase (NTR), an enzyme involved in nitro reduction, dehydrogenation, and flavin fragmentation, is overexpressed in hypoxia, which occurs when the median concentration of tumor tissue is between 0% and 4% in a given area [87,88,89]. The creation of an efficient assay tool is crucial for the investigation of NTR activity in biological systems. NIR fluorescent probes to detect NTR in vitro and in vivo have recently been reported by several groups.

In 2013, Zhang et al. described an NIR fluorescent probe **26** for NTR detection that consists of a cyanine dye as the NIR fluorophore and the nitro-group as the recognition unit (Figure 3) [70]. When Probe **26** was exposed to NTR containing NADH as an electron donor, there was a dramatic increase in fluorescence intensity at 708 nm, resulting in a 50-fold fluorescence enhancement in 20 min. When the NTR activates the probe, intramolecular self-elimination and electron rearrangement occur, resulting in fluorescence release (Figure 4B). The selectivity studies reveal that the probe is unaffected by several biological reductants, implying that it may be employed for NTR detection under normal physiological settings. In another investigation, Hu et al. created a cyanine-based NIR fluorescent probe **27** for detecting NTR activity (Figure 3) [71]. The probe emits NIR fluorescence after the nitro group is reduced to an amino group and has a high selectivity toward NTR. For the first time, it was also employed to undertake fairly quickly detection and comparative evaluation of NTR activity in ESKAPE infections. Zhang et al. published an NIR fluorescent probe **28** for the selective detection of NTR in 2018. It combines with NTR to produce the fluorophore NIR-MAO, which selectively accumulates in hypoxia mitochondria and emits at 710 nm. It is subsequently changed to NIR-MAOH in autolysosomes containing acidified mitochondria and glows at 675 nm [72] (Figure 4C). Finally, the results of cell imaging by the probe showed that the fluorescence ratio decreased after reoxygenation, and the level of mitophagy was further increased after hypoxia was induced again, suggesting that mitophagy may be a self-protective process that adapts cells to hypoxia.

Wu et al. described an NTR-activated NIR probe **29** for the in situ localization of breast cancer metastases using NIR fluorescence imaging and photoacoustic imaging. (Figure 3) [73]. The amphiphilic molecular nature of the probes allows them to be readily assembled into nanoparticles in an aqueous environment. Because of the nanoaggregates’ loosely packed structure, when the probe reacts with NTR, the electron-withdrawing nitro unit is quickly reduced to an amino group. This causes an intramolecular self-elimination reaction to release the fluorophore, which is followed by AIE fluorescence and photoacoustic signal output. Using dual model imaging and the modified probe **29**, tumor metastasis suppression during cyclophosphamide chemotherapy could be tracked in real-time.

In 2019, Peng et al. constructed an NTR-activated NIR fluorescent probe **30** for the effective use of photosensitizers against solid tumors (Figure 3) [74]. The structure of the probe mainly includes the iodinated heptamethine cyanine as an NIR photosensitizer group, and the 4-nitrobenzyl chloroformate as a classical NTR trigger group. When the probe interacted with NTR, the fluorescence intensity at 830 nm steadily decreased with time and NTR concentration. The existence of “PET-induced emission quenching” and “PET blocking” processes resulted in a fluorescence quantum yield of 10.8% for probe **30**, much higher than 0.4% for Icy-NH_2_ (Figure 4D). The results show that the photothermal conversion efficiency of Icy-NH_2_ is 35.2%, which is more than twice that of ICG (16%). HeLa cell imaging revealed that probe **30** fluoresced strongly in normoxic (21 percent O_2_) settings and weakly in hypoxic (0.1 percent O_2_) conditions, demonstrating that under hypoxic conditions, the endogenous endogenous NTR can activate the probe, which in turn releases Icy-NH_2_. The PDT and PTT effects of probe **30** could be resolutely switched for the tumor photoablation under 808 nm light irradiation. Intriguingly, the experimental results unmistakably demonstrate that under 808 nm light irradiation, the probe produces heat under hypoxic conditions for effective hypoxia oxygen photothermal destruction, in addition to intracellular singlet oxygen in the normoxic region for effective PDT treatment. Finally, in vivo solid tumor therapy experiments show that the probe produces excellent therapeutic effects. In another investigation, peng and co-workers reported a novel NTR-activatable NIR photoredox catalysis fluorescent probe **31** (Figure 3) [75]. The probe shows high selectivity toward NTR, and it reacts with NTR to generate Se-NH_2_ through intramolecular self-elimination, and the maximum emission is at 710 nm (Figure 4E). With an O_2_-independent mode of action, Se-NH_2_ is able to start NIR photoredox catalysis and control cellular electron pools. In a mouse model, effective tumor-specific photocatalytic eradication (TGI: 95%) was accomplished using this novel approach. High-resolution tumor identification (SBR: 33.6), superior biocompatibility, and safety are further benefits.

In 2020, zhao and colleagues created an NTR-activatable and mitochondrial-targetable NIR fluorescent probe **32** by connecting *p*-nitrobenzyl thioether and triphenylphosphine to the BODIPY. (Figure 3) [76]. NTR can trigger the recognition group to form aminobenzyl sulfide and undergo a 1,6-elimination reaction to release the NIR fluorophore. Given the mitochondrial-targetable features, ratiometric fluorescence (F_560nm_/F_713nm_) and NIR emission luminescence (λ_em_ = 713nm) in response NTR, the probe provides high-precision imaging of mitochondria in cancer cells by measuring nitroreductase activity in real time. Very recently, Kwan and co-workers reported a NIR cancer-targeting fluorescent probe **33** for NTR detection (Figure 3) [77]. The probe is mainly constructed based on BODIPY as the main skeleton, and includes folate as the cancer cell targeting group and *p*-nitrobenzyl as the NTR triggering unit. A remarkable fluorescence enhancement at 730 nm was observed for probe **33** in the presence of NTR. The probes was quickly taken up by folate receptor-positive CT26 cancer cells and produced greater NIR signals under hypoxic circumstances than the control probe, according to in vitro investigations. After tail vein injection, the probe discriminately accumulated in the solid tumor location, with a maximum rise in fluorescence intensity occurring within one day. Probe **33** is a promising fluorescent “off-on” probe with future uses due to its tumor-targeting capabilities and imaging capacity of particular activation of hypoxic tumor areas.

### 4.3. NIR Fluorescent Probes for NAD(P)H:Quinone Oxidoreductase 1

Under oxidative stress, NAD(P)H:quinone oxidoreductase 1 (NQO1) detoxifies quinones to hydroquinone by a two-electron reduction reaction using the reducing cofactor NAD(P)H. According to the study, tumors had higher levels of NQO1 than normal tissues did. NQO1 is a good target for early cancer diagnosis, prognosis, and therapy due to its association with cancer [90]. As a result, there is an increasing interest in creating efficient tools to investigate NQO1 activity in biological systems, and there are more reports of NIR fluorescent probes based on NQO1 activation.

In 2017, McCarley et al. developed a NIR fluorescent probe **34** for detection and visualization of endogenous intracellular NQO1 activity (Figure 3) [78]. The probe consists of three parts, namely the tricarbocyanine TCy as the reporter, the trimethyl locked quinone propionic acid as a highly selective hNQO1 substrate and the 2-mercaptoethanol as a linker (Figure 4F). The probe was specifically reduced by hNQO1, and then the fluorophore Tcy was released through continuous intramolecular ring closure and accompanied by fluorescence release (λ_em_ = 755 nm), and the final fluorescence multiple increased by 193-fold in 20 min. The characteristic target-to-background signal provided by this probe can detect and differentiate human cancer cells with different levels of NQO1 activity, including cells that experience different microenvironments due to their location in multicellular tumor mimics. Local application of a diluted solution of the probe in a mouse xenograft model identifies human ovarian cancer tumors as small as 0.5 mm in size.

Using similar design strategies, probe **35**, **36**, and **37** were reported by different groups, all of which exhibited high selectivity for NQO1 (Figure 3) [79,80,81]. Both probes **35** and **36** manifested ratiometric fluorescence changes during the detection of NQO1, and their maximum fluorescence emission spectra are *F*_595nm_/_725nm_ and *F*_555nm_/_650nm_, respectively. With the help of imaging by the probe **35**, the researchers found that the levels of enzyme activity in organ tissues varied widely. It is interesting to note that the probe **36** has two-photon imaging capabilities, which will eventually make its in vivo imaging abilities much more convenient. Probe **37** exhibits environment-insensitive emission behavior and hence has potential for researching the enzyme in diverse biological systems. Under physiological circumstances, the probe demonstrated remarkable responsiveness, sensitivity, and optical performance to NQO1, and it was capable of detecting endogenous NQO1 activity in live cells with an extremely high fluorescent enhancement (108-fold). Notably, it was effectively utilized to high-contrast imaging of tumor-bearing mice, with a maximal fluorescence response achieved in 10 min.

### 4.4. NIR Fluorescent Probes for Cyclooxygenase-2

Cyclooxygenase 2 (COX-2) catalyzes the important biosynthesis of arachidonic acid to various prostaglandins during various inflammatory responses, which are critical in inflammatory processes and are found to be upregulated early in tumorigenesis, and play an important role in the expansion and development of cancer [91]. As a consequence, COX-2 is a crucial biomarker for cancer detection and inflammation.

In 2014, peng and coworkers reported a NIR fluorescent probe **38** for COX-2 detection in cancer cells, which consists of the nile blue as NIR fluorophore, the indomethacin (IMC) as recognition unit and the hexanediamine as a linker (Figure 3) [82]. Computational results show that the fluorescence is quenched due to the PET effect between Nile blue and IMC, and the specific binding of the COX-2 structure and IMC leads to the release of fluorescence. The probe can be used for rapid, sensitive and quantitative flow cytometry screening of cancer cells. At the same time, due to the excellent NIR fluorescence optical imaging properties of the probe technology, this offers great potential for non-invasive detection of tumors in vivo. Probe **39** utilizes a similar design principle to **38** to realize the switch of fluorescence (Figure 3) [83]. With the help of the probe, the experimental results showed that with the aggravation of pulmonary fibrosis, the level of COX-2 increased in the early stage and decreased in the late stage. In addition, up-regulation of COX-2 levels was effective in reducing the severity of pulmonary fibrosis. Therefore, it is a fast and convenient imaging tool with great potential in predicting the early stages of pulmonary fibrosis and assessing treatment effects.

### 4.5. NIR Fluorescent Probes for Thioredoxin Reductase

Thioredoxin reductase (TrxR) is a key antioxidant enzyme in the thioredoxin (Trx) system that regulates intracellular redox homeostasis. TrxR takes electrons from NADPH and transfers them to Trx, and the reduced txr reacts with downstream proteins, thereby restoring the homeostasis of intracellular maintenance [92,93,94]. Thus, TrxR activity is critical for maintaining appropriate oxidative stress. It is critical to develop real-time detection systems to understand the role of TrxR in various diseases.

Our group reported a series of fluorescent probes (TRFS series) based on disulfide five-membered rings that can be used for real-time detection of intracellular TxrR activity, resulting in an overall high selectivity of the probe molecules for TrxR [95,96,97,98]. Probe **40** is the fluorescent probe with the longest reported emission wavelength for the detection of TrxR. (Figure 3) [84]. The probe demonstrated strong TrxR selectivity in the experimental setup, particularly by excluding the effects of some significant amounts of reducing agents present in the cell, such as GSH. TrxR reduces the probe, which is followed by the formation of sulfhydryl group to produce a fluorophore with strong NIR fluorescence (Figure 4G). Finally, TrxR activity in live cells was quickly detected using it with success.

## 5. NIR Fluorescent Probes for Transferases

Transferases are enzymes that catalyze the nucleophilic substitution process that transfers functional groups from one substrate to another. Irregularities in this enzyme family have been linked to a variety of disorders, including Alzheimer’s disease, amyotrophic lateral sclerosis (ALS), and schizophrenia. As a result, having an efficient detection method is critical for understanding their involvement in biochemical transitions. Figure 5 displays sample examples of NIR fluorescent transferase probes that have recently been published, and Table 3 summarizes the findings of photophysical characteristics and kinetics experiments.

**Figure 5 molecules-27-05922-f005:**
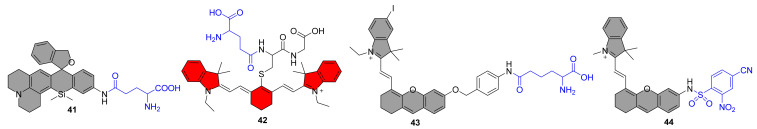
Structures of NIR Fluorescent Probes **41**–**44**.

### 5.1. NIR Fluorescent Probes for γ-Glutamyl Transferase

γ-Glutamyltransferase (GGT) is an enzyme present in cell membranes that catalyzes the transfer of the γ-glutamyl bond of glutathione (GSH) to the receptor. GGT is crucial to the γ-glutamyl cycle and is involved in glutathione’s breakdown. Changes in GGT levels have been linked to a number of illnesses, including cancer, cardiovascular disease, and liver disease [103]. To effectively monitor GGT activity in tissues, effective and quick techniques for evaluating GGT activation fluorescent probes are needed.

Urano et al. reported an activatable NIR fluorescent probe **41**, and as expected, the probe’s fluorescence emission was red-shifted to the NIR region at 662 nm, with a more than 145-fold increase after the addition of GGT (Figure 5) [99]. The pH titration test of the probe demonstrated that it was fully closed at the neutral buffer with a p*K*_cycl_ of 4.8, making it suitable for use in solution and cellular assays. The probe detected tiny tumor nodules in the mesentery, indicating that it can be used in clinical settings. Li et al. reported an example of a NIR ratiometric fluorescent probe **42** for the detection of GGT shortly after (Figure 5) [100]. The glutathione unit was chosen as the substrate for targeting GGT in this probe and was coupled to a near-infrared fluorophore (Cy-N). The probe emits strong fluorescence at 805 nm, and when GGT is added, the γ-glutamyl chain degrades enzymatically, releasing negatively charged amino groups that undergo nucleophilic substitution reactions to produce amino substitution products, which may then undergo charge transfer from indocyanine to nucleophilic nitrogen to produce Cy-N, which emits at 640 nm (Figure 6A). The probe may be used to monitor GGT activity in a variety of biological materials, including blood, cells, tissues, and in vivo, and the results indicate that it could be used to identify cancer early. Recently, Kong et al. created a near-infrared (NIR) fluorescent probe **43** with a therapeutic function that can combine fluorescent endogenous GGT detection for tumor diagnostics and NIR imaging-guided tumor photodynamic treatment (Figure 5) [101]. The probe fluoresces at 718 nm in response to GGT with great selectivity and sensitivity (0.014 U/L), allowing both in vitro and in vivo monitoring of GGT activity. Furthermore, when exposed to NIR light, this GGT-responsive probe produces more ROS, which promotes tumor cell death and inhibits tumor development. Finally, multifunctional fluorescence probes with complementary benefits have considerable potential for tumor identification and therapy in complex organisms.

**Figure 6 molecules-27-05922-f006:**
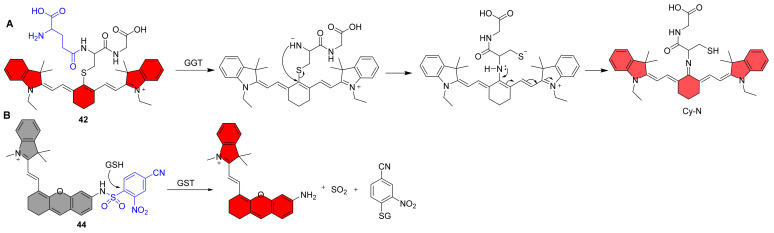
Recognition mechanism of probe **42** and **44**.

### 5.2. NIR Fluorescent Probes for Glutathione Transferases

Glutathione Transferases (GSTs) are enzymes that catalyze a variety of metabolic pathways during biotransformation by employing the cofactor GSH, which is required for external detoxification and cellular oxidative stress. GST activity in tumors is critical for treatment resistance and cell survival, and GST overexpression has been seen in a range of cancers [104,105]. As a result, there is an urgent need to research a validated approach for determining GST activity in biological systems.

Chen et al. reported a NIR fluorescent probe **44** for the detection of GSTP1, which is the subclasses of cytosolic GSTs (Figure 5) [102]. Probe **44** consists of an electron-deficient GST recognizing group and the NIR fluorophore. Mechanistically, the probe is activated by GSH via GST-catalyzed cleavage of the sulfonamide bond to form the GS conjugate, sulfur dioxide, and the NIR fluorophore (Figure 6B). Upon incubation with GSTP1/GSH, the new GST probe demonstrates over 100-fold fluorescence increase and good selectivity over other possible interference. Additionally, it has been demonstrated that the probe can monitor endogenous GST in A549 cells. Finally, because the probe emits in the near-infrared range, it only causes little photodamage to biological material. Deeper tissue can be penetrated, especially when the fluorescent background interference is mild. It is ideal for in vivo imaging investigations due to these benefits.

## 6. Summary and Perspective

NIR fluorescent probes based on enzyme activation have become a valuable tool for viewing physiological and pathological processes in cells and in vivo. Meanwhile, because of their unique characteristics of strong biocompatibility, facile cell metabolism, and high sensitivity, they have been frequently utilized in fluorescence-mediated diagnostics. Based on a study of past work, we conduct a comprehensive evaluation of the design methodologies and research development of enzyme-responsive NIR fluorescent probes in this publication. According to the data we reviewed, NIR probes are becoming increasingly prevalent for in vivo imaging research. Furthermore, enzyme-activated NIR fluorescent probes, as compared to pure dyes for photodynamic or photoacoustic therapy, have demonstrated greater potential to regulate the differential treatment of distinct tissue locations, which makes them appealing in this sector.

Despite significant progress in this field, a number of problems remain, and significant efforts are required to encourage and grow new advancements. (1) When the content of the object to be detected is very low or the effective reaction form of the detected substance exists for a short length of time, the selectivity and sensitivity of the probe molecule for enzyme detection should be increased. Furthermore, fluorescent probes have a difficult time selectively recognizing different isoforms of the same enzyme family, such as methionine sulfoxide reductase, carboxylesterase, cytochrome p450, nucleotide pyrophosphatase, nucleotide phosphodiesterase, and cyclooxygenase, among others. According to research, the development of novel recognition sites using molecular docking or inhibitor bionics is advantageous for the production of highly selective fluorescence probes. (2) According to our results, almost all enzyme-activated NIR fluorescent probes have activated fluorescence emission in the NIR I region, as opposed to fluorophores in the NIR II region (1000–1700 nm), which have the advantages of stronger tissue penetration and a higher signal-to-noise ratio and will be widely used in diagnostic and medical fields. However, there have been few publications on the creation of enzyme-activated fluorescent probes utilizing NIR II region fluorophores, limiting in situ studies of carcinogenesis, development, and proliferation. (3) Fluorescence imaging alone gives limited useful information, and fluorescence imaging paired with photoacoustic, magnetic resonance, or ^19^F imaging may be a more potent multimodal imaging strategy to properly image enzymes in complicated biological systems. (4) The vast majority of probe molecules employed in our summarized study were used to identify tumor-associated bioactive species. Tumorigenesis is produced by intricate molecular control and overexpression of many enzymes at the molecular level. Unfortunately, most fluorescent probes created to date can only discriminate between normal and malignant cells and are incapable of identifying particular tumor cell types. As a result, a multi-target enzyme-activated fluorescence probe based on a cascade reaction would give an approach for identifying cancer cells and tumors in vivo. (5) NIR fluorescent dyes generally require a large conjugated chemical structure, resulting in inferior water solubility, reduced quantum yields in aqueous solutions, and highly challenging fluorescence signal detection in cells. When NIR probes are loaded into cells alongside large amounts of organic solvents, the intracellular environment is likely to be interfered. Modifying the probe backbone with water-soluble chemical groups, such as carboxyl and sulfonic acid groups, may be a viable solution for this problem. Meanwhile, the overall complexity and expense of NIR probe chemical production are significant in the case short wavelength probes (<650 nm). (6) The chemical structural stability of the fluorescent probe is critical for the accuracy of the analytical results. However, there are very few tests on this aspect in the works summarized in this review. It is worth investigating if the chemical structure of the probe is cleaved over time under physiological settings, or whether altering pH has an influence on the chemical structure in the presence of the probe alone. (7) Most of the described work gives a hazy study of the exact mechanism of action of probes and analytes, and no specific verification findings, such as HPLC or LCMS data, is provided. It appears that just changing the fluorescence signal cannot satisfy the rigorous threshold for evidence of scientific study. Finally, we believe that the examples and debates in this review will help with the creation of new platforms for novel NIR fluorescence probes triggered by enzymes, as well as provide new possibilities for the clinical diagnosis and treatment of linked disorders.

## Figures and Tables

**Figure 1 molecules-27-05922-f001:**
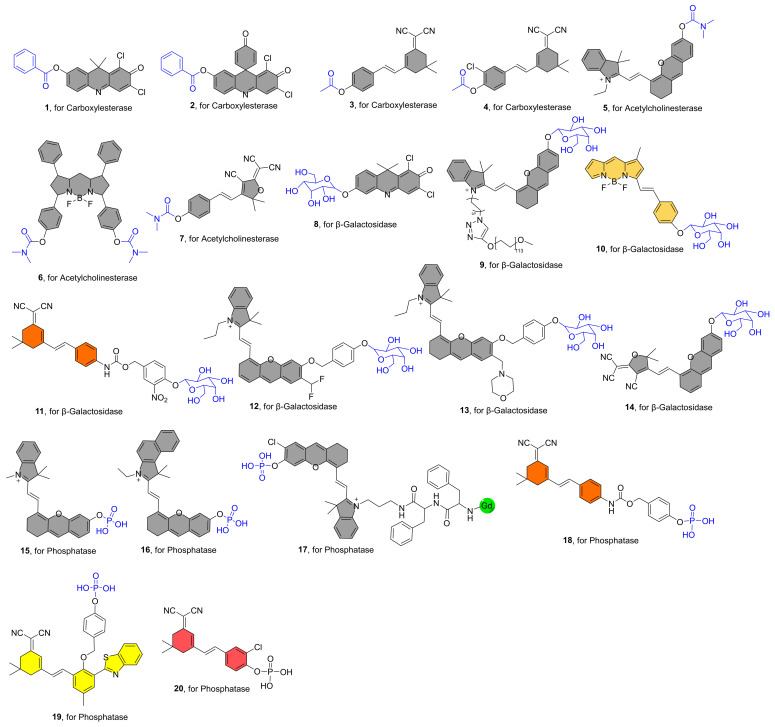
Structures of NIR Fluorescent Probes **1**–**20** (The gray filling of the chemical structure shows that the fluorophore molecule’s fluorescence has been quenched, while the other colors indicate that the probe molecule exhibits corresponding fluorescence signals when it does not respond to the analyte, the following Figure 2, Figure 3, Figure 4, Figure 5 and Figure 6 are the same).

**Table 1 molecules-27-05922-t001:** Properties of NIR fluorescent Probes **1**−**20**.

Probe Number	Original Name	Trigger	Solvent *^a^*	Signal Transduction Mode	Excitation, Maximal Emission (nm)	LOD *^b^*	K_m_ *^c^*	Localized Organelles	Application	Ref.
**1**	DDAB	CE	PBS	“turn-on”	F646Ex, 662Em	0.07 ug/mL	1.927 ± 0.31 uM	n.a.	Live cells and mice	[38]
**2**	DSAB	CE	PBS	“turn-on”	F630Ex, 678Em	0.03 ug/mL	6.55 ± 0.49 uM	n.a.	Live cells and high-throughput screening	[39]
**3**	CE-1	CE	PBS/DMSO = 4:1, *v*/*v*	“turn-on”	F535Ex, 665Em	2.76 × 10^−3^ U/mL	n.a.	n.a.	Live cells and mice	[40]
**4**	ZM-1	CE	PBS	“turn-on”	F495Ex, 665Em	0.287 × 10^−3^ U/mL	5.4 uM	n.a.	Live cells and mice	[41]
**5**	CyN	AChE	PBS	“turn-on”	F670Ex, 700Em	0.1173 U/mL	25.36 uM	n.a.	Live cells and Zebrafish	[42]
**6**	BD-AChE	AChE	HEPES	“turn-on”	F710Ex, 740Em	0.21 U/mL	85 uM	n.a.	Live cells and mice aging models	[43]
**7**	EW3	AChE	PBS	“turn-on”	F560Ex, 690Em	0.17 U/mL	n.a.	n.a.	Live cells and Zebrafish	[44]
**8**	DDAOG	β-gal	PBS	“turn-on”	F636Ex, 659Em	n.a.	n.a.	n.a.	Live cells and mice	[45]
**9**	CyGal-P	β-gal	HEPES	“turn-on”	F675Ex, 720Em	n.a.	48.3 uM	n.a.	Live cells and photothermal therapy in mice	[46]
**10**	BODIPY-βgal	β-gal	PBS/DMSO = 1:1, *v*/*v*	ratiometric	F560Ex, 575Em/F660Ex, 730Em	4.6 U/L	n.a.	n.a.	Live cells and mice	[47]
**11**	TMG	β-gal	PBS	ratiometric	F410Ex, 580Em/F445Ex, 660Em	0.86 U/L	24.04 uM	n.a.	Live cells	[48]
**12**	NIR-BG2	β-gal	PBS	“turn-on”	F596Ex, 709Em	n.a.	9.3 uM	n.a.	Live cells and mice	[49]
**13**	Lyso-Gal	β-gal	PBS/DMSO = 4:1, *v*/*v*	“turn-on”	F690Ex, ~720Em	22 U/L	n.a.	lysosome	Live cells	[50]
**14**	DMC-βgal	β-gal	PBS/DMSO = 7:3, *v*/*v*	“turn-on”	F725Ex, 770Em	0.298 U/L	n.a.	n.a.	Live cells and mice	[51]
**15**	NALP	ALP	Tris-HCl/DMSO = 19:1, *v*/*v*,	“turn-on”	F680Ex, 706Em	0.28 U/L	52.45 uM	n.a.	Live cells and mice	[52]
**16**	CyP	ALP	Tris-HCl	“turn-on”	F690Ex, 738Em	3 U/L	9.32 uM	n.a.	Live cells and mice	[53]
**17**	PCyFF-Gd	ALP	Tris	“turn-on”	F680Ex, 710Em	0.017 U/L	13.14 uM	n.a.	Fluorescence/MRI bimodal imaging in mice	[54]
**18**	APT	ALP	Tris-HCl	ratiometric	F410Ex, 580Em/F445Ex, 650Em	0.89 U/L	1.64 uM	n.a.	Live cells and Zebrafish	[55]
**19**	HP	ALP	Tris-HCl	ratiometric	F398Ex, 556Em/F423Ex, 689Em	3.98 U/L	2.93 uM	n.a.	Live cells and Zebrafish	[56]
**20**	SWJT-3	ALP	PBS/DMSO = 4:1, *v*/*v*	ratiometric	F405Ex, 590Em/F405Ex, 670Em	0.87 U/L	8.89 uM	n.a.	Live cells and mice	[57]

*^a^* Detailed composition in the original literature. *^b^* Limit of detection (LOD). *^c^* Michaelis−Menten kinetics (K_m_) values. n.a = not available.

**Table 2 molecules-27-05922-t002:** Properties of NIR fluorescent Probes **20**−**40**.

Probe Number	Original Name	Trigger	Solventt *^a^*	Signal Transduction Mode	Excitation, Maximal Emission (nm)	LOD *^b^*	K_m_ *^c^*	Localized Organelles	Application	Ref.
**21**	MAORed-1	MAO	HEPES	ratiometric	F437Ex, 550Em/F437Ex, 664Em	1.2 ug/mL	270 uM for MAO-B	n.a.	Live cells	[65]
**22**	MitoCy-NH_2_	MAO-B	HEPES	ratiometric	F730Ex, 803Em/F650Ex, 750Em	n.a.	10.13 ± 0.28 uM	n.a.	Live cells and mice aging models	[66]
**23**	DHMP2	MAO-A	PBS/DMSO = 19:1, *v*/*v*	“turn-on”	F680Ex, 710Em	13.0 ng/mL	9.4 uM	mitochondria	Live cells, zebrafish and tumor-bearing mice	[67]
**24**	Rma-1	MAO-A	PBS	“turn-on”	F680Ex, 708Em	4.5 ng/mL	30.3 ± 3.8 uM	n.a.	Live cells, zebrafish and mice	[68]
**25**	KXS-M2	MAO-A	PBS/DMSO = 4:1, *v*/*v*	“turn-on”	F500Ex, 670Em	9.8 ng/mL	12.1 uM	n.a.	Live cells and Zebrafish	[69]
**26**	Probe (5)	NTR	PBS	“turn-on”	F590Ex, 708Em	n.a.	301.8 uM	n.a.	Detection of NTR activity under physiological conditions (tube)	[70]
**27**	Probe 1	NTR	PBS	“turn-on”	F658Ex, 699Em	8.4 ng/mL	7.21 uM	n.a.	Bacterial	[71]
**28**	NIR-HMA	NTR	PBS/DMSO = 19:1, *v*/*v*	“turn-on”	F670Ex, 710Em	n.a.	n.a.	n.a.	Live cells	[72]
**29**	Q-NO_2_	NTR	PBS	“turn-on”	F680Ex, 780Em	0.052 ug/mL	n.a.	n.a.	Live cells and mice	[73]
**30**	Icy-NBF	NTR	HEPES/DMSO = 7:3, *v*/*v*	“turn-on”	F813Ex, 830Em	n.a.	n.a.	n.a.	Photodynamic and photothermal therapies	[74]
**31**	Se-NO_2_	NTR	PBS/DMSO = 9:1, *v*/*v*	“turn-on”	F660Ex, 710Em	n.a.	51.8 uM	n.a.	Photocatalytic theranostics	[75]
**32**	BOD-TPP	NTR	Tris-HCl/DMSO = 3:1, *v*/*v*	ratiometric	F495Ex, 565Em/F650Ex, 713Em	0.0168 ug/mL	33.7 uM	mitochondria	Live cells	[76]
**33**	fol-BODIPY	NTR	DMEM containing 10% FBS	“turn-on”	F642Ex, 730Em	1.52 ng/mL	n.a.	n.a.	Live cells and mice	[77]
**34**	Q3STCy	NQO1	PBS	“turn-on”	F605Ex, 755Em	n.a.	1.1 ±0.5 uM	n.a.	Live cells and mice	[78]
**35**	QRP3	NQO1	PBS	“turn-on”	F520Ex, 595Em/F650Ex, 725Em	139 μg/mL	n.a.	n.a.	Live cells and mice tissues	[79]
**36**	Py^+^BC_NQO1	NQO1	Tris-HCl	ratiometric	F360Ex, 555Em/F470Ex, 650Em	4.99 μg/mL	1.13 uM	n.a.	Live cells	[80]
**37**	S-QCy7-NQO1	NQO1	PBS	“turn-on”	F640Ex, 740Em	0.05 μg/mL	n.a.	n.a.	Live cells and mice	[81]
**38**	Niblue-C6-IMC	COX-2	n.a.	“turn-on”	F630Ex, 670Em	n.a.	n.a.	n.a.	Live cells and mice	[82]
**39**	Cy-COX	COX-2	Tris-HCl	“turn-on”	F650Ex, 770Em	11 ng/mL	n.a.	n.a.	Live cells and mice	[83]
**40**	TRFS-red	TrxR	Tris-EDTA	“turn-on”	F615Ex, 661Em	n.a.	n.a.	n.a.	Live cells	[84]

*^a^* Detailed composition in the original literature. *^b^* Limit of detection (LOD). *^c^* Michaelis−Menten kinetics (K_m_) values. n.a = not available.

**Table 3 molecules-27-05922-t003:** Properties of NIR fluorescent Probes **41**−**44**.

Probe Number	Original Name	Trigger	Solvent *^a^*	Signal Transduction Mode	Excitation, Maximal Emission (nm)	LOD *^b^*	K_m_ *^c^*	Localized Organelles	Application	Ref.
**41**	gGlu-HMJSiR	GGT	PBS	“turn-on”	F580Ex, 661Em	n.a.	n.a.	n.a.	Live cells and mice	[99]
**42**	Cy-GSH	GGT	PBS	ratiometric	F730Ex, 805Em/F540Ex, 640Em	0.03 U/L	9.87 uM	n.a.	Live cells and mice	[100]
**43**	CyI-Glu	GGT	PBS/DMSO = 9:1, *v*/*v*	“turn-on”	F690Ex, 718Em	0.014 U/L	n.a.	n.a.	Photodynamic therapy in mice	[101]
**44**	GSTC	GST	PBS	“turn-on”	F670Ex, 695Em	0.037 ug/mL	0.39 uM	n.a.	Live cells and mice	[102]

*^a^* Detailed composition in the original literature. *^b^* Limit of detection (LOD). *^c^* Michaelis−Menten kinetics (K_m_) values. n.a = not available.

## Data Availability

Not applicable.

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
