# Peer review of "Recent Progress on NIR Fluorescent Probes for Enzymes"

_molecules, 2022, doi:10.3390/molecules27185922_

Round 1

Reviewer 1 Report

The authors introduce the recent research results of fluorescent probes for enzymes, which are worthy of publication before solving the following problem:

1. The authors should summarize the contents at the beginning of the paper, making it easy to find specific enzyme probes.

2. which organelles are those probes located in?  Please note in the tables.

Author Response

Reviewer 1

The authors introduce the recent research results of fluorescent probes for enzymes, which are worthy of publication before solving the following problem:

We appreciate the positive comments from the reviewer.

  1. The authors should summarize the contents at the beginning of the paper, making it easy to find specific enzyme probes.

Response: We have done this in Page1&2 (The yellow highlight).

  1. which organelles are those probes located in? Please note in the tables.

Response: We have done this in Tables (Page 31-33, the yellow highlight).

Reviewer 2 Report

This manuscript reviews NIR fluorescent probes with various photophysical characteristics and their use in detecting enzyme activity under biological settings. The authors briefly discussed the examples of typical probes for selected class of enzymes that may be used to observed enzyme activity in biological systems.

The purpose of the review is clear and the examples are rather well described. There are, however, some shortcomings, which needs to be addressed by the authors, as listed below.

The authors should expand section considering the current problems and limitations related to the use of organic probes. In addition to the shortcomings mentioned in the “Summary and Perspective”, attention should also be paid to the lack of solubility of most probe in the water and the negative impact of the organic solvents used in the analysis. Organic cosolvent may result in cytotoxicity or interfere with the assays, for example, by scavenging the analyte of interest. A serious and often overlooked limitation in the use of probes is the lack of identification of probes conversion products in biological systems (e.g. using HPLC or LCMS techniques), but only the measurement of the generated fluorescence signal without determining the exact mechanism of its formation. The lack of knowledge about the chemical structure of the emerging fluorescent species does not allow to determine whether the increase in fluorescence intensity results from the reaction with a given analyte or whether it is independent of this reaction. Therefore, detailed characterization of the in vivo product modification of the probes is crucial to understanding the role that individual enzymes play in biological processes. The NIR probes are often characterized by low fluorescence quantum yield in water, which makes it difficult to register a weak fluorescent signal in biological systems, and thus it limits the use of such probes as tools for determining enzyme activity. The above limitations on the use of probes should be considered in the manuscript. Moreover, the title of the article should be changed as the authors only discuss probes for the detection of selected classes of enzymes.

Author Response

Reviewer 2

This manuscript reviews NIR fluorescent probes with various photophysical characteristics and their use in detecting enzyme activity under biological settings. The authors briefly discussed the examples of typical probes for selected class of enzymes that may be used to observed enzyme activity in biological systems.

The purpose of the review is clear and the examples are rather well described. There are, however, some shortcomings, which needs to be addressed by the authors, as listed below.

The authors should expand section considering the current problems and limitations related to the use of organic probes. In addition to the shortcomings mentioned in the “Summary and Perspective”, attention should also be paid to the lack of solubility of most probe in the water and the negative impact of the organic solvents used in the analysis. Organic cosolvent may result in cytotoxicity or interfere with the assays, for example, by scavenging the analyte of interest. A serious and often overlooked limitation in the use of probes is the lack of identification of probes conversion products in biological systems (e.g. using HPLC or LCMS techniques), but only the measurement of the generated fluorescence signal without determining the exact mechanism of its formation. The lack of knowledge about the chemical structure of the emerging fluorescent species does not allow to determine whether the increase in fluorescence intensity results from the reaction with a given analyte or whether it is independent of this reaction. Therefore, detailed characterization of the in vivo product modification of the probes is crucial to understanding the role that individual enzymes play in biological processes. The NIR probes are often characterized by low fluorescence quantum yield in water, which makes it difficult to register a weak fluorescent signal in biological systems, and thus it limits the use of such probes as tools for determining enzyme activity. The above limitations on the use of probes should be considered in the manuscript. Moreover, the title of the article should be changed as the authors only discuss probes for the detection of selected classes of enzymes.

Response: We appreciate the positive comments from the reviewer. The points raised by the reviewer are critical. We apologize that these issues were not addressed in the original manuscript. We have addressed a discussion of these issues in the revised manuscript in Page 18 (The yellow and green highlight). In this review, we chose several different types of enzymes to discuss their relevant NIR fluorescent probes, and their recognition mechanisms are of diverse sorts. Furthermore, distinct enzymes are differentiated by the disease models that they are connected with. We'd like to preserve the original title. However, for the convenience of readers, we have included a table of contents in the text (Page 1 &2, the yellow highlight).

Reviewer 3 Report

The authors describe an article entitled “Recent Progress on NIR Fluorescent Probes for Enzymes”. The topic of the manuscript is interesting, and the manuscript constitutes an interesting review concerning the development of NIR dyes for biological applications.

The work is well-written and a well-constructed introduction has been established by the authors. Sufficient spectra and figures are included in the manuscript for comprehension and clarity. Interesting and convincing results are also presented in this work. Overall, I think that this is a manuscript that I recommend for publication after inclusion of minor revisions.

1) A table presenting the pros and cons of NIR Probe vs other probes should be presented in the introduction section.

2) What about the cost of NIR dyes compared to others fluorescent probes ?

3) More colors should be added in the different schemes in order to render them more attractive.

4) What about the stability of these probes ? This point is not examined.

5) A Table listing the emission wavelengths of all probes should be added.

Author Response

Reviewer 3

The authors describe an article entitled “Recent Progress on NIR Fluorescent Probes for Enzymes”. The topic of the manuscript is interesting, and the manuscript constitutes an interesting review concerning the development of NIR dyes for biological applications.

The work is well-written and a well-constructed introduction has been established by the authors. Sufficient spectra and figures are included in the manuscript for comprehension and clarity. Interesting and convincing results are also presented in this work. Overall, I think that this is a manuscript that I recommend for publication after inclusion of minor revisions.

We appreciate the positive comments from the reviewer.

  1. A table presenting the pros and cons of NIR Probe vs other probes should be presented in the introduction section.

Response: The major advantage of NIR probes vs other probes is that their emission band is greater than 650 nm, and there are numerous literature publications on the benefits of this. At the same time, we have an overview in the text's introduction (Page 3, the yellow highlight). However, because the conjugated chemical structure of NIR probes is substantially bigger than other fluorescent probe types of chemical structures, water solubility is relatively poor, and the complexity and expense of synthesis are higher. This section of the discussion has been added to the revised text (Page 18, the yellow highlight).

  1. What about the cost of NIR dyes compared to others fluorescent probes?

Response: The cost of NIR dyes is generally higher than other fluorescent probes, but it is still within an acceptable and manageable range.

  1. More colors should be added in the different schemes in order to render them more attractive.

Response: We appreciate that the reviewer gives an excellent recommendation for improving the article's readability. We added more colors in figures of chemical structure (Figure 1-6).

  1. What about the stability of these probes? This point is not examined.

Response: The reviewer raised an important question. The stability of the fluorescent probe is critical for the accuracy of the analytical results. However, there are very few tests on this aspect in the works summarized in this review. Thus we apologize for not covering this issue in the article, but we added some words to further discuss this issue in the major body of the  revised Summary and Perspective part (Page 18, the purple highlight).

  1. A Table listing the emission wavelengths of all probes should be added.

Response: We have addressed the emission wavelengths of all probes in the original manuscript’s Tables.

Round 2

Reviewer 2 Report

It can be accepted for publication with this form.